# RVO-MIS: Robust Visual Odometry for Minimally Invasive Surgery

**Zhuo Wang**[1] (iD)                                                    ZWANG570@ARIZONA.EDU
**Chiang-Heng Chien**[2] (iD)                              CHIANG-HENG_CHIEN@BROWN.EDU
**Eungjoo Lee**[1,3] (iD)                                        EUNGJOOLEE@ARIZONA.EDU

[1] *Department of Electrical and Computer Engineering, University of Arizona, Tucson, AZ, USA*

[2] *School of Engineering, Brown University, Providence, RI, USA*

[3] *Department of Ophthalmology and Vision Science, University of Arizona, Tucson, AZ, USA*

**Editors:** Accepted for publication at MIDL 2026

## Abstract

Visual odometry (VO) in minimally invasive surgery (MIS) scenarios plays a crucial role in current and future endoscopic surgical intervention assistance systems. However, MIS environments pose severely challenging situations for typical VO algorithms due to textureless scenes, the presence of surgical instruments, light reflections, flowing blood and organ deformation, *etc.* Classic VO methods adopt a smooth motion prior to generate an initial guess for camera pose and then refine it through minimizing reprojection errors. Recent deep learning methods incorporate learned depths and estimate camera poses through minimizing photometric residuals. These approaches, however, lack robustness in estimation due to abrupt motion change and unpredictable illumination changes commonly seen in MIS environments. In this paper, we present RVO-MIS, a robust VO framework in MIS by first integrating SIFT and LightGlue for reliable feature correspondences, and then solving a sequence of absolute camera poses under a M-estimator sample consensus (MSAC) scheme. By advocating the absolute-pose-first formulation to prioritize geometric consistency and robustness, our approach decouples the camera motion tracking from smooth motion prior, photometric consistency, learned depths, *etc.* Through evaluations on the SCARED and EndoSLAM datasets, RVO-MIS demonstrates consistently accurate camera pose estimations. In challenging MIS situations where many methods fail or become inaccurate, RVO-MIS excels in both camera trajectory completion rate and accuracy. Code is publicly available at https://github.com/vsi-lab/RVOMIS.git.

**Keywords:** Visual Odometry, Minimally Invasive Surgery, Feature-based Tracking

## 1. Introduction

Accurate camera pose estimation is an important component of navigation and guidance in minimally invasive surgery (MIS). This surgical navigation system, capable of tracking the laparoscope and displaying the spatial relationship between the laparoscope and surrounding anatomical structures, can effectively reduce the risk of critical organ damage caused by excessive contact during surgery while enhancing the surgeon's spatial awareness. Compared to marker-based navigation systems, vision-based approaches demonstrate higher efficiency as they do not interrupt surgical procedures and have the potential to achieve real-time navigation (Ye et al., 2025; Liu et al., 2022). The objective is to accurately estimate the 6 degrees of freedom (DoF) camera motion from a sequence of ordered monocular images in MIS scenarios, as illustrated in Figure 1.

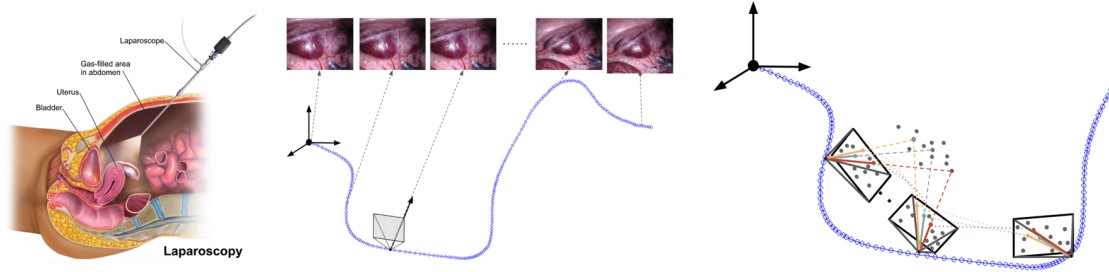

(a) Laparoscopic surgery setup     (b) 6-DoF motion estimation and trajectory     (c) 3D triangulation and pose optimization

Figure 1: Illustration of the RVO-MIS framework in an MIS environment. (a) A typical laparoscopic surgery setup. Image adapted from (Blausen.com staff, 2014). (b) Visual odometry estimates 6-DoF camera motion, generating a continuous patient-internal trajectory. (c) 3D points arising from keyframes serve as geometric anchors for robust absolute pose estimation in a MSAC scheme, which is the core of the proposed RVO-MIS.

Most existing camera pose estimation methods are based on visual odometry (VO) or simultaneous localization and mapping (SLAM) frameworks, such as ORB-SLAM2 (Mur-Artal and Tardós, 2017), and its variations (Rodríguez et al., 2021) designated for non-rigid surgical scenes. These approaches typically start with sparse or semi-sparse feature point correspondences, and recovers the camera motion by first suggesting an initial camera pose based on the assumption that the camera motion is temporally constant, and then refine the initial guess through minimizing reprojection errors. Other approaches, specifically the deep learning-based family, have witnessed the deployment of learned depths (Zhong et al., 2024; Bhat et al., 2023; Zhang et al., 2025; Yang et al., 2024) from monocular images to the traditional VO or SLAM methods, *e.g.*, Endo-Depth-and-Motion (Recasens et al., 2021), EndoSLAM (Ozyoruk et al., 2021), *etc.* In particular, the learned depths complement the need for sparse features which are hard to be detected in textureless scenes, and estimates camera poses either through minimizing photometric residuals or PoseNet architectures (Kendall et al., 2015) trained with depth-guided constraints.

Despite these advances, VO methods relying on estimated depth images inevitably suffer from errors introduced by depth estimation, and the depth errors are easily propagated to the errors in estimating camera poses. The assumption on photometric consistency also breaks easily in the MIS environment when light is unpredictable in a sense that it is reflected on organs, liquid, surgical instruments, *etc.* On the other hand, methods based on ORB-SLAM2 (Mur-Artal and Tardós, 2017) framework face significant challenges in complex MIS scenarios. Specifically, its assumption on smooth temporal motion prior is violated when camera moves drastically, a common scenario in the MIS environment. In addition, even though there are ample features recognized from MIS images, constructing correspondences based on feature descriptor similarity is unstable, often leading to insufficient feature correspondences for a reliable pose estimation (Chien et al., 2024).

In this work, we propose RVO-MIS, a robust VO method in MIS that combines the sub-pixel accuracy of classical SIFT (Lowe, 2004) features with the deep learning-based matching, LightGlue (Lindenberger et al., 2023), for reliable feature correspondences, integrated with absolute pose estimation for camera pose estimation. To reconcile the rigid-body

assumption of absolute pose from perspective-3-points (P3P) with non-rigid MIS deformations, we integrate M-estimator sample consensus (MSAC) (Torr and Zisserman, 2000) to robustly filter out features detected from deforming tissues, ensuring stable camera motion tracking on quasi-rigid background structures. The core philosophy of the proposed method is to decouple camera pose estimation process from smooth-motion priors, photometric consistency, or joint optimization over pose, depth, and scene structure, which are the primary building blocks of the existing methods, but are often violated in the challenging MIS environments. We showed that, by prioritizing geometric consistency and robustness supported by integrating a deep learning-based feature matcher over temporal smoothness, multi-frame optimization, and strong scene modeling constraints, the proposed design offers a complementary alternative to the competing methods. In addition, in contrast to the computationally intensive approaches presented in (Recasens et al., 2021; Ozyoruk et al., 2021; Yang et al., 2023; Hayoz et al., 2023) that require extensive GPU resources for scene-specific training of depth estimation networks, our framework employs deep learning only for feature extraction and matching. By leveraging pre-trained weights with demonstrated generalization capabilities (Mackutė et al., 2024), our method achieves robust feature detection and matching performance without requiring additional scene-specific fine-tuning. The contributions are outlined as follows:

- **Absolute-pose-first VO formulation:** RVO-MIS recovers camera motion by robustly estimating a sequence of absolute poses, rather than through temporally coupled minimization of reprojection errors or photometric residuals. This removes reliance on smooth motion assumption, and leaves refining the pose through minimizing reprojection errors as a lightweight optimization rather than the primary operation.

- **MSAC-based pose estimation from robust feature detection and matching as the core estimator:** the proposed framework integrates the sub-pixel accuracy of classical SIFT features with the robust deep learning-based matching (LightGlue). This combination effectively addresses challenging MIS conditions such as textureless regions and specular reflections without requiring scene-specific fine-tuning.

- **Empirical validation through challenging datasets:** we demonstrate *state-of-the-art performance* on SCARED (Allan et al., 2021) and EndoSLAM (Ozyoruk et al., 2021) datasets, both in accuracy and trajectory completion rate. An ablation study reveals that inlier ratio significantly increases by using SIFT in conjunction with Light-Glue compared to the similarity-based approach, whereas absolute pose estimation in a MSAC scheme contributes to robust and accurate camera motion recovery.

## 2. Related Work

**SLAM Methods:** SLAM enables real-time tracking and mapping, a critical capability for MIS. SAGE (Liu et al., 2022) integrates learned priors with factor graph optimization to ensure robust reconstruction in textureless, illumination-varying environments. Another framework (Wu et al., 2022) combines medical bag-of-words with Poisson reconstruction, generating dense, detailed 3D models from sparse outputs. Addressing visual failure, ArthroSLAM (Marmol et al., 2018) utilizes a dynamically weighted extended Kalman filter

for continuous multi-sensor localization. Finally, feature-based methods (Deng et al., 2023b) significantly improve tracking performance by combining K-means with SuperPoint (DeTone et al., 2018) for enhanced feature extraction. To address the generalization gap in deep learning-based SLAM, BodySLAM (Manni et al., 2024) achieves cross-domain generalization without fine-tuning by combining CycleGAN-based pose estimation with zero-shot depth prediction. Recently, Endo-2DTAM (Huang et al., 2025) leverages 2D Gaussian Splatting and surface normal-aware tracking to overcome multi-view inconsistencies, enabling high fidelity, geometrically accurate reconstruction.

**VO Methods:** A hybrid approach (Song et al., 2021) integrating deep learning networks and geometric features was implemented in this scenario. Region classification and a two-stage pose refinement procedure are the two main components of this novel approach. It uses a Siamese network architecture (Bromley et al., 1993) and two identical PoseNet models (Kendall et al., 2015) to assess the similarity between the test image and its collected region. Pose is obtained via triangulation using region information. This data-efficient approach outperforms pure deep learning or geometry methods. Furthermore, sensor fusion is highlighted as a solution to the inherent scale drift and ambiguity of conventional monocular VO. In this context, EndoVMFuseNet (Turan et al., 2017) uses a recurrent CNN to fuse 6DoF visual and 5DoF magnetic data without synchronization. Its energy reduction method integrates dense photometric alignment with sparse optical flow features. While sensor fusion enhances data richness, DPVO (Teed et al., 2023) maximizes efficiency by tracking sparse patches instead of dense flow. It combines a recurrent update operator with differentiable bundle adjustment, achieving the robustness of dense methods with significantly reduced computational and memory costs.

**Depth Estimation Methods:** (Chen et al., 2019) propose a cGAN-based framework using a U-Net generator for depth estimation. By enforcing geometric fidelity through adversarial training and fusing estimates within ElasticFusion (Whelan et al., 2016), it achieves robust real-time reconstruction. More recently, (Yang et al., 2023) introduces a geometry-aware framework based on MultiDepth (Watson et al., 2021). By employing a composite loss function targeting gradient and normal consistency, this approach significantly enhances geometric fidelity for complex anatomical features, achieving state-of-the-art performance on the EndoSLAM dataset (Ozyoruk et al., 2021).

## 3. Method

### 3.1. Overview

The core innovation of RVO-MIS lies in promoting geometric consistency in estimating camera poses in a robust feature correspondence construction and MSAC-based absolute pose estimation framework, Figure 2. Specifically, the input is a sequence of ordered monocular images. An initialization is in charge of giving rise to a set of 3D cloud of points to provide an inherent scale for the resultant trajectory. These 3D points are used to associate with the 2D point correspondences for estimating absolute poses of subsequent frames. When the 2D-3D correspondences are scarce, a frame is elevated to a keyframe, and the process continues which in the end returns a complete trajectory. Notably, RVO-MIS is free from assuming smooth camera motion and photometric consistency, and does not rely on scene model-based constraints and learned depths.

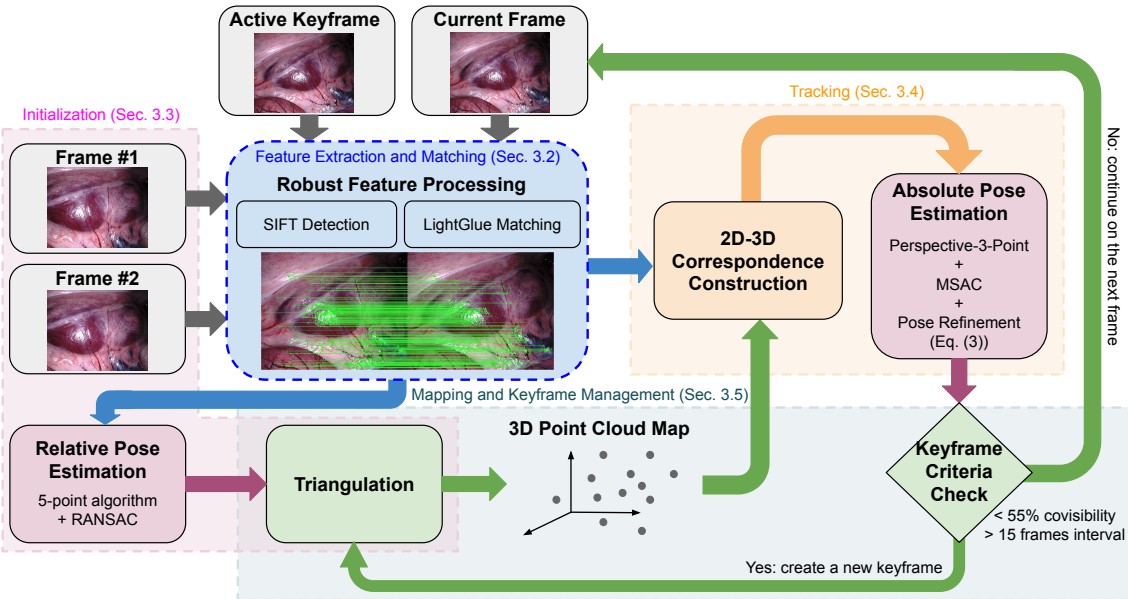

Figure 2: The proposed VO framework begins with feature extraction using SIFT and robust feature matching via LightGlue (Section 3.2). During the initialization stage, relative pose estimation and triangulation are performed to reconstruct an initial set of 3D points (Section 3.3). For subsequent frames, 2D features detected in the current image are associated with 3D points from the active keyframe, enabling absolute pose estimation via a P3P solver within an MSAC scheme, followed by a lightweight least-squares refinement (Section 3.4). Keyframes are selectively introduced to incrementally enrich the 3D point cloud, facilitating reliable absolute pose estimation for later frames (Section 3.5).

**Notations:** Let $\Gamma_{w,k}$ be a $k$-th 3D point in the world coordinate, $K$ be the camera calibration matrix, $\mathcal{R}_i$ and $\mathcal{T}_i$ be the estimated absolute rotation matrix and translation vector of camera $i$, respectively. A 2D feature point $\gamma_k$ with depth $\rho_k$ gives rise to the $k$-th 3D point in the camera coordinate $\Gamma_{i,k} = \rho_k \gamma_k$ which relates the 3D point $\Gamma_w$ in the world coordinate by

$$\Gamma_{i,k} = \mathcal{R}_i \Gamma_{w,k} + \mathcal{T}_i. \tag{1}$$

Denote $\gamma_{im,k}$ as the $k$-th image point in pixels so that $\gamma_{im,k} = K\gamma_k$.

## 3.2. Feature Extraction and Matching

As shown in (Mackutė et al., 2024), a diverse set of features was analyzed for endoluminal navigation. It revealed that modern learning-based features such as SuperPoint (DeTone et al., 2018) provide denser features compared to conventional features such as SIFT (Lowe, 2004), ORB (Rublee et al., 2011), *etc*, they remain suboptimal in terms of accuracy and reliability. In contrast, SIFT is the most accurate and reliable features. These findings align with our empirical evidence, and thus we adopt SIFT in the proposed framework to prioritize accuracy and reliability.

To ensure high quality candidates and computational efficiency, we rank-ordered the features and pick the top 80% of features ($\rho_{rank} = 0.8$) based on their response scores. During the matching phase, we initially employed the VLFeat (Vedaldi and Fulkerson, 2010) for constructing SIFT correspondences based on SIFT descriptor similarity, with additional filters of Lowe's ratio test and bidirectional consistency check. However, our prior experiments showed that a significant proportion of SIFT correspondences are ambiguous and non-veridical, exacerbating the subsequent pose estimation. Alternatively, we adopted LightGlue (Lindenberger et al., 2023), a state-of-the-art learning-based feature matcher. It leverages a graph neural network (GNN) to jointly reason about feature correspondences, incorporating both local appearance and geometric consistency. In this case, SIFT correspondences become ample and reliable, which is supported by our ablation study, Section 4.

### 3.3. Initialization

After constructing a set of SIFT correspondences from the first two frames, relative pose is estimated via 5-point algorithm (Nistér, 2004) under a classic RANSAC (Fischler and Bolles, 1981) scheme. A 2D-2D correspondence is treated as an inlier supporting the relative pose hypothesis when their distance to the corresponding epipolar line is below 2 pixels. The inlier supports of the best relative pose hypothesis are used to triangulate to their 3D counterparts, forming a cloud of 3D points under the world (first frame) coordinate. Thus, the estimate of absolute poses of subsequent frames using the constructed 3D points is inherently referenced to the world coordinate, so that the inherent scale remains consistent throughout the entire trajectory. Once the initialization is complete, the second frame is elevated to be an active keyframe as a reference image which gives rise to the 3D points.

### 3.4. Tracking

Upon reconstructing 3D points from inlier 2D feature matches, we identify which 2D features in the keyframe have valid 3D correspondences. By matching 2D features between the current frame and the keyframe, the association of observed 2D points in the current frame and the reconstructed 3D points (referred to as co-visible 3D points) can be found. These 2D-3D correspondences enable an absolute pose estimation of the current frame via solving a P3P minimal problem. In the presence of incorrect associations, MSAC (Torr and Zisserman, 2000) is adopted for a robust absolute pose estimation with a maximum of 3000 iterations. Each absolute pose hypothesis is supported by an inlier 2D-3D point pair if its reprojection error is less than 2 pixels. This MSAC-based absolute pose estimation is the main building block of recovering the camera motion in RVO-MIS. To further boost the estimation accuracy, the estimated absolute pose of camera $i$ is refined by minimizing an energy function $E(\mathcal{R}_i, \mathcal{T}_i)$ defined as a function of the absolute rotation $\mathcal{R}_i$ and absolute translation $\mathcal{T}_i$ in the form of a sum of reprojection errors, *i.e.*,

$$E(\mathcal{R}_i, \mathcal{T}_i) = \frac{1}{2} \sum_{k=1}^{N} \left\| \gamma_{im,k} - \frac{K(\mathcal{R}_i \Gamma_{w,k} + \mathcal{T}_i)}{e_3^T K(\mathcal{R}_i \Gamma_{w,k} + \mathcal{T}_i)} \right\|^2, \tag{2}$$

so that the refined absolute pose $(\mathcal{R}_i^*, \mathcal{T}_i^*)$ is

$$(\mathcal{R}_i^*, \mathcal{T}_i^*) = \operatorname*{argmin}_{\mathcal{R}_i, \mathcal{T}_i} E(\mathcal{R}_i, \mathcal{T}_i). \tag{3}$$

Minimizing $E(\mathcal{R}_i, \mathcal{T}_i)$ with respect to $(\mathcal{R}_i, \mathcal{T}_i)$ is done by the Levenberg-Marquardt algorithm. Note that the rotation matrix $\mathcal{R}_i$ is parameterized by the three Euler angles, and thus there are six unknowns in total, *i.e.*, three for rotation and three for translation.

### 3.5. Mapping and Keyframe Management

Our keyframe update strategy triggers under two conditions: *(i)* when fewer than 55% of the current frame's co-visible 3D landmarks originate from the active keyframe, or *(ii)* when exceeding 15 frames since the last keyframe insertion. As a new keyframe is inserted, we triangulate newly observed point features between the current frame and the active keyframe, *i.e.*, the inlier 2D matches whose 3D counterparts are absent. Triangulating 2D point feature matches from two views requires the relative transformation which can be easily achieved by coordinate transformation of the current camera pose and the keyframe camera pose. Specifically, let $(\mathcal{R}_c, \mathcal{T}_c)$ and $(\mathcal{R}_k, \mathcal{T}_k)$ be the absolute poses of the current frame and the keyframe, respectively; the goal is to find the relative pose $(\mathcal{R}_{kc}, \mathcal{T}_{kc})$ so that a point $\Gamma_c$ under the current camera coordinate is transformed to a point $\Gamma_k$ under the keyframe camera coordinate by $\Gamma_k = \mathcal{R}_{kc}\Gamma_c + \mathcal{T}_{kc}$. From Equation 1, we have

$$\begin{cases} \Gamma_c = \mathcal{R}_c\Gamma_w + \mathcal{T}_c \\ \Gamma_k = \mathcal{R}_k\Gamma_w + \mathcal{T}_k, \end{cases} \tag{4}$$

where for simplicity we omit the index for the point. Now, $\Gamma_w$ can be isolated in the first vector equation of Equation 4 as

$$\Gamma_w = \mathcal{R}_c^T \left( \Gamma_c - \mathcal{T}_c \right), \tag{5}$$

which can be plugged to the second vector equation of Equation 4, giving

$$\Gamma_k = \mathcal{R}_k\mathcal{R}_c^T \left( \Gamma_c - \mathcal{T}_c \right) + \mathcal{T}_k = \mathcal{R}_k\mathcal{R}_c^T\Gamma_c + \mathcal{T}_k - \mathcal{R}_k\mathcal{R}_c^T\mathcal{T}_c. \tag{6}$$

Thus, the relative pose $(\mathcal{R}_{kc}, \mathcal{T}_{kc})$ is

$$\begin{cases} \mathcal{R}_{kc} = \mathcal{R}_k\mathcal{R}_c^T \\ \mathcal{T}_{kc} = \mathcal{T}_k - \mathcal{R}_k\mathcal{R}_c^T\mathcal{T}_c. \end{cases} \tag{7}$$

which provides epipolar constraint across the two frames as the essential matrix $E_{kc} = [\mathcal{T}_{kc}]_\times\mathcal{R}_{kc}$ can be easily found. This constraint is adopted to pick 2D-2D correspondences whose 3D counterparts are absent that satisfy epipolar geometry from thresholding the Sampson error (Terekhov and Larsson, 2023). Those that pass the epipolar geometry constraints are triangulated to form a group of *new* 3D points and are transformed to the world coordinate in order to keep the entire cloud of 3D points in the same coordinate system. This ensures continuous map expansion while maintaining consistent inherent scale through geometric verification. A complete RVO-MIS pipeline can be found in Appendix A.

## 4. Experiments

**Dataset:** The proposed method is evaluated on multiple sequences of the stereo correspondence and reconstruction of endoscopic data (SCARED) (Allan et al., 2021) dataset. Specifically, these sequences were chosen to cover a diverse set of conditions: two of them exhibit

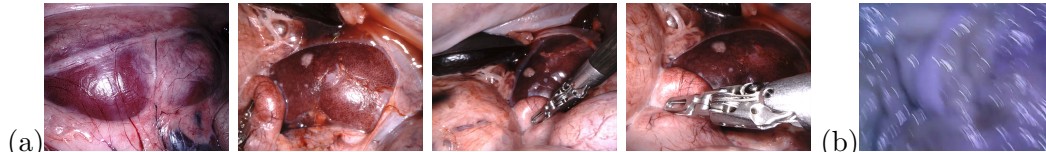

Figure 3: Example images of (a) the chosen four sequences (ordered from left to right) from the SCARED dataset, exhibiting scenes with light reflections, flowing blood, presence of surgical instruments, *etc*; (b) the chosen sequence of EndoSLAM dataset with abrupt camera motion, attributing to blurry images and skipping frames.

illumination variations and flowing blood, and another two are challenging by the presence of surgical instruments and textureless areas (Figure 3(a)). The EndoSLAM dataset (Ozyoruk et al., 2021) is also used. Recognizing that the EndoSLAM dataset has a frame-skipping problem (Deng et al., 2023a), namely, the sequence of images is not consecutive, often giving non-overlap in scene between adjacent images. This often leads to excessive large errors for all methods after evaluation, undermining the validity of performance metrics on this dataset. Nevertheless, EndoSLAM provides situations where camera moves drastically, creating small overlaps in scenes between adjacent frames and attributing to blurry images (Figure 3(b)). This challenging situation is adopted in this paper, specifically the small intestine sequence, to test the failure handling strategies of all methods.

**Baselines:** The proposed RVO-MIS is compared against several methods, ranging from classic geometry-based approaches to the recent learning-based contemporaries. In particular, ORB-SLAM2 (Mur-Artal and Tardós, 2017) and SD-DefSLAM (Rodríguez et al., 2021) are representatives for classic non-learning based methods, while BodySLAM (Manni et al., 2024), Endo-2DTAM (Huang et al., 2025), DPVO (Teed et al., 2023), and Endo-Depth (Recasens et al., 2021) are recent learning-based methods. ORB-SLAM2 and DPVO are designed for generic scenes, while the rest targets MIS applications. Throughout our experiments, we use default settings for all competing methods, *i.e.*, for learning-based approaches their pretrained weights are used. In addition, native image resolutions from the datasets are fed to all methods without any preprocessing, *e.g.*, down-sampling, *etc.*

**Evaluation Metrics:** For quantitative evaluations, standard metrics (Sturm et al., 2012) are adopted. Specifically, (Umeyama, 1991)'s method is first applied for a global trajectory alignment, where the inherent scale ambiguity of monocular methods is accounted for by a similarity transformation in Sim(3). Absolute trajectory error (ATE) of each aligned estimated pose with respect to the corresponding ground-truth pose is used, from which aggregated errors over the entire trajectory is described by means of root-mean-square error (RMSE). Throughout the paper, $\text{ATE}(\mathcal{T})$ and $\text{ATE}(\mathcal{R})$ respectively represent the RMSE of ATE for the translation (in centimeters) and the rotation (in degrees) parts. For the experiments on the EndoSLAM dataset, we additionally use Trajectory Completion Rate (Chien et al., 2024), namely, the ratio of the number of images a method estimates their poses until failure over the total number of images in the sequence, to demonstrate the performance of the competing methods in situations with a high chance of failure.

**Quantitative Results on SCARED:** Tables 1 and 2 respectively summarize the performance comparisons on the SCARED dataset through *global alignment*, *i.e.*, the geometry of the estimated trajectory is best aligned with the geometry of the ground-truth trajectory, and *origin alignment*, *i.e.*, the first frame is identical across all methods. RVO-MIS demon-

Table 1: Quantitative comparison of ATE (RMSE) using *global alignment* on four representative sequences from the SCARED dataset. **Bold**: best. underlined: second best. ("M": Monocular, "S": Stereo, "-": Tracking Failure)

| Method | Sequence 1 | | Sequence 2 | | Sequence 3 | | Sequence 4 | |
|---|---|---|---|---|---|---|---|---|
| | ATE ($\mathcal{T}$) | ATE ($\mathcal{R}$) | ATE ($\mathcal{T}$) | ATE ($\mathcal{R}$) | ATE ($\mathcal{T}$) | ATE ($\mathcal{R}$) | ATE ($\mathcal{T}$) | ATE ($\mathcal{R}$) |
| ORB-SLAM2 (M)  (Mur-Artal and Tardós, 2017) | 1.6543 | 3.0534 | 0.9912 | 3.0798 | 9.6247 | 2.8640 | 4.4825 | 3.0183 |
| ORB-SLAM2 (S)  (Mur-Artal and Tardós, 2017) | 1.3091 | 0.3511 | 1.3623 | 3.0303 | 9.7053 | 3.0089 | 4.3626 | 2.9151 |
| SD-DefSLAM (M)  (Rodríguez et al., 2021) | 1.0036 | 0.3247 | 1.7788 | 2.0058 | 6.2579 | 0.3455 | 3.5653 | 0.1761 |
| BodySLAM (M)  (Manni et al., 2024) | 0.4504 | 0.1991 | 0.4447 | 0.1851 | - | - | 8.2481 | 0.7986 |
| Endo-2DTAM (M)  (Huang et al., 2025) | 2.3290 | 2.2410 | 2.3187 | 1.1858 | 6.3792 | 1.9767 | 5.1394 | 2.3166 |
| DPVO (M)  (Teed et al., 2023) | 0.3326 | 0.2466 | 0.3902 | 0.1763 | **2.8271** | 0.4607 | 0.9269 | 0.4405 |
| EndoDepth (M)  (Recasens et al., 2021) | 1.1748 | 0.3688 | 1.4863 | 1.7176 | 9.2599 | 0.8804 | 4.2948 | 0.3484 |
| **RVO-MIS (M) (Proposed)** | **0.2970** | **0.0523** | **0.3574** | **0.1302** | 4.0381 | **0.1261** | **0.6822** | **0.0548** |

Table 2: Quantitative comparison of ATE (RMSE) using *origin alignment* on four representative sequences from the SCARED dataset. **Bold**: best. underlined: second best. ("M": Monocular, "S": Stereo, "-": Tracking Failure)

| Method | Sequence 1 | | Sequence 2 | | Sequence 3 | | Sequence 4 | |
|---|---|---|---|---|---|---|---|---|
| | ATE ($\mathcal{T}$) | ATE ($\mathcal{R}$) | ATE ($\mathcal{T}$) | ATE ($\mathcal{R}$) | ATE ($\mathcal{T}$) | ATE ($\mathcal{R}$) | ATE ($\mathcal{T}$) | ATE ($\mathcal{R}$) |
| ORB-SLAM2 (M)  (Mur-Artal and Tardós, 2017) | 8.3241 | 0.1259 | 98.5070 | 1.4628 | 234.2704 | 17.9475 | 128.9915 | 1.9776 |
| ORB-SLAM2 (S)  (Mur-Artal and Tardós, 2017) | 3.4019 | 0.0524 | 5.9156 | 0.1176 | 27.8263 | 0.5553 | 17.2080 | 0.2732 |
| SD-DefSLAM (M)  (Rodríguez et al., 2021) | 3.1928 | 0.0391 | 1.3217 | 0.0319 | 8.5289 | 0.1610 | 3.8524 | 0.6624 |
| BodySLAM (M)  (Manni et al., 2024) | 2.7290 | 0.4290 | 0.8442 | 0.2093 | - | - | 11.2834 | 0.4392 |
| Endo-2DTAM (M)  (Huang et al., 2025) | 9.6275 | 2.2411 | 3.0723 | 1.1861 | 13.2626 | 2.1547 | 8.1023 | 2.3157 |
| DPVO (M)  (Teed et al., 2023) | 1.1042 | 0.4124 | 0.5583 | 0.1252 | **3.5819** | 0.5303 | 2.3902 | 0.5074 |
| EndoDepth (M)  (Recasens et al., 2021) | 3.0702 | 0.0534 | 1.8933 | 0.0386 | 10.4578 | 0.2422 | 6.4815 | 0.1013 |
| **RVO-MIS (M) (Proposed)** | **0.7963** | **0.0179** | **0.5528** | **0.0236** | 4.7195 | **0.0830** | **1.3573** | **0.0237** |

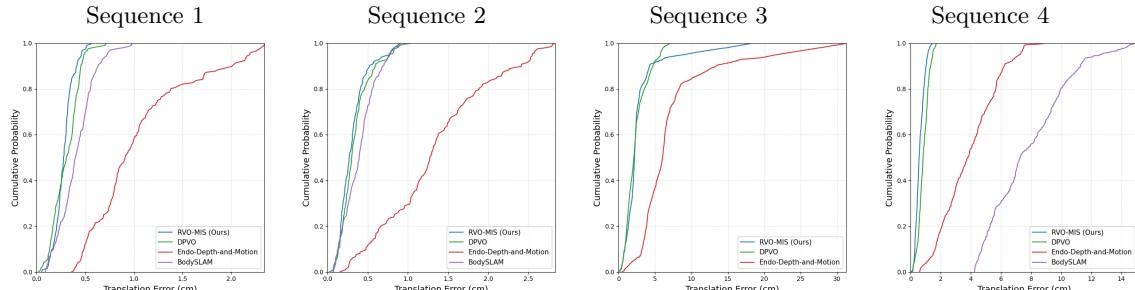

Figure 4: CDF of the translation ATE across representative sequences. The curves illustrate the cumulative distribution of per-frame errors. In sequences 1, 2, and 4 of the SCARED dataset, RVO-MIS (blue) demonstrates superior accuracy. In sequence 3, while DPVO achieves lower overall errors, our method maintains a consistent error distribution for the majority of frames, highlighting the varying challenges posed by different surgical scenes.

strates superior performances for most of the sequences in both alignments. Although the accuracy reported in the table is averaged over the entire sequence, the cumulative density functions (CDFs) of ATE errors in Figure 4 show that the error does not exhibit sudden jumps typically associated with tracking loss. Rather, it increases gradually and in a largely monotonic fashion, reflecting slow drift accumulation rather than catastrophic failure. This consistent robustness highlights the effectiveness of the RVO-MIS framework for MIS navigation, particularly in dynamic environments where stable tracking is critical. Additional detailed results can be found in Appendix B.

Table 3: Quantitative evaluation on the Small Intestine sequence of the EndoSLAM dataset using global alignment (GA) and origin alignment (OA). **Bold**: the best. Underlined: the second best.

| Method | Fail? | Avg. Traj. Compl. (%) | GA-ATE (RMSE) | | OA-ATE (RMSE) | |
|---|---|---|---|---|---|---|
| | | | $\mathcal{T}$ | $\mathcal{R}$ | $\mathcal{T}$ | $\mathcal{R}$ |
| ORB-SLAM2 (Mur-Artal and Tardós, 2017) | Yes | 8.67% | 8.71 | 11.88 | 14.22 | 9.60 |
| SD-DefSLAM (Rodríguez et al., 2021) | Yes | 16.98% | 7.45 | 5.54 | 7.27 | 3.53 |
| BodySLAM (Manni et al., 2024) | Yes | 28.82% | 3.01 | 3.74 | 3.99 | 2.52 |
| Endo-2DTAM (Huang et al., 2025) | No | 100% | 2.41 | 2.30 | 3.46 | 2.21 |
| DPVO (Teed et al., 2023) | No | 100% | 2.25 | 0.89 | 3.93 | **0.49** |
| EndoDepth (Recasens et al., 2021) | Yes | 38.24% | 4.97 | 3.31 | 5.43 | 3.94 |
| **RVO-MIS (Proposed)** | **No** | 100% | **2.02** | **0.26** | **3.32** | 0.73 |

The inferior performances from the competing methods are studied as follows. *(i)* The reliance on smooth motion prior as an initial guess for camera poses in ORB-SLAM2 (Mur-Artal and Tardós, 2017) and SD-DefSLAM (Rodríguez et al., 2021) is often violated in the MIS environment where camera motion is abrupt and erratic. Since minimizing reprojection errors is not a convex optimization process, a poor initial guess would converge to a local optima, leading to inaccurate estimation. The strong dependency on temporal coherence is also the case in Endo-2DTAM (Huang et al., 2025). *(ii)* The reliance on a strong parametric prior for modeling the deformable shape for estimating the pose in BodySLAM (Manni et al., 2024) acts like an over-constrained optimization, which is often biased especially in the unpredictable, changing MIS environment. *(iii)* Photometric consistency assumption, primarily adopted by DPVO (Teed et al., 2023) and EndoDepth (Recasens et al., 2021), is easily collapsed in the presence of significant light changes in MIS scenarios.

**Quantitative Results on EndoSLAM:** Table 3 shows the performances of RVO-MIS compared to the existing approaches on the challenging EndoSLAM dataset. Notably, ORB-SLAM2, SD-DefSLAM, EndoDepth, and BodySLAM all fail to track the camera motion towards the end of the sequence. This is typically because of poor pose estimations in some early frames encountering blurry images imposed by drastic camera motion change that leads to failure in the later frames, as supported by their diminished accuracy. On the other hand, Endo-2DTAM and DPVO are designed to keep estimating camera poses, despite the estimation being poor. This enforces the completion of the trajectory but with lower accuracy compared the proposed approach.

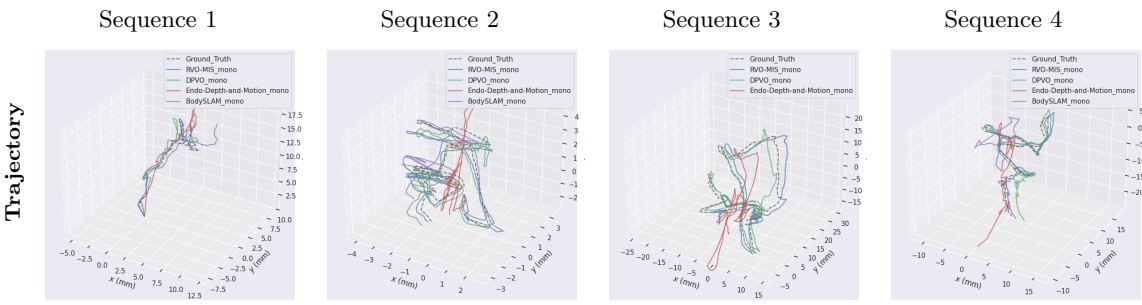

Figure 5: Trajectories (origin alignment) comparison for the representative sequences in the SCARED dataset. Our proposed method (blue) closely tracks the Ground Truth (black, dashed), outperforming established baselines.

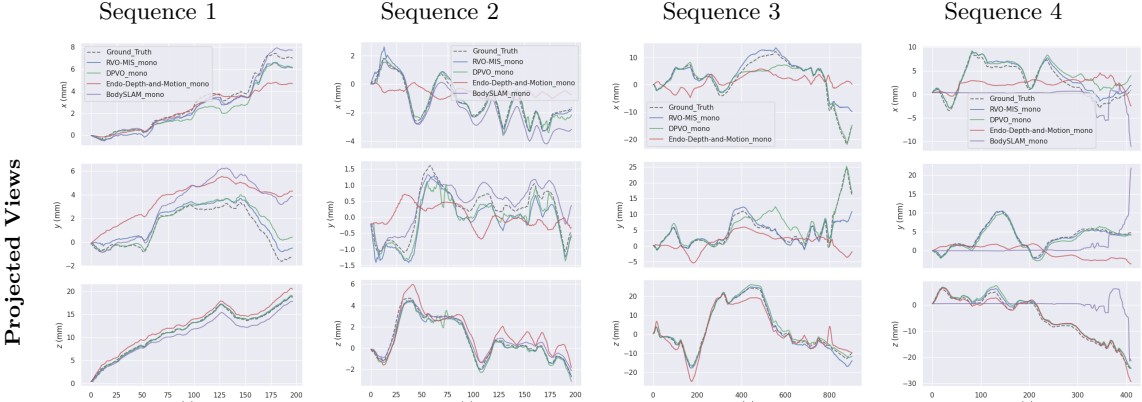

Figure 6: Projected trajectory views $(x, y, z$ components) for the corresponding sequences in Figure 5, clearly visualizing the direction and magnitude of motion drift over time.

**Qualitative Results:** A qualitative assessment of the trajectories using the SCARED dataset is presented in Figure 5 compared to the baselines, with a clearer view for each dimension presented in Figure 6. These trajectories are visualized with origin alignment, *i.e.*, the first frame is identical across everyone, facilitating an observation of motion drift. From Figure 6, it is clear that the proposed method performs better than others by a wide margin, except for some cases DPVO is comparative with us.

**Ablation Study:** To examine the effectiveness of the feature matching method as well as the use of geometric verification combinations, an ablation study is conducted on both SCARED and EndoSLAM datasets, Table 4, in terms of average inlier rate and average trajectory completion. Evidently, LightGlue delivers around 18% improvements in inlier ratio over the traditional descriptor similarity based method (VLFeat). This is also reflected in an inlier ratio box plot across the selected sequences of the SCARED dataset, Figure 7. Although MSAC offers higher inlier ratio over RANSAC by a small margin, it elevates significant robustness in estimating absolute poses. Note that although VLFeat+RANSAC has better accuracy compared to LightGlue+RANSAC, the former is measured across shorter trajectory length than the latter.

**Run Time:** RVO-MIS was implemented in Python to avoid cross-language interoperability overhead and leverage parallel computing capabilities. Experiments were performed on a high performance server with dual AMD EPYC 9354 32-Core Processors (4 CPU cores for this study) and a single NVIDIA RTX A6000 GPU (48GB VRAM). For a runtime breakdown experiment, when running on a total of 197 frames, feature matching accounted for

Table 4: Ablation study of varying matching and geometric verification combinations. We report the average Inlier Ratio (%), Average Trajectory Completion Rate (%), and Average ATE (RMSE) using global alignment averaged across SCARED and EndoSLAM datasets.

| Method | Avg. Inlier (%) ↑ | Avg. Traj. Compl. (%) ↑ | ATE ($\mathcal{T}$) ↓ | ATE ($\mathcal{R}$) ↓ |
|---|---|---|---|---|
| VLFeat+RANSAC | 38.7722 | 82.6652 | 3.0613 | 0.8025 |
| VLFeat+MSAC | 37.9260 | 82.8458 | 2.7302 | 0.3540 |
| LightGlue+RANSAC | 52.1536 | 100.0000 | 5.3533 | 1.4494 |
| **LightGlue+MSAC (Ours)** | **52.6790** | **100.0000** | **2.0311** | **0.3023** |

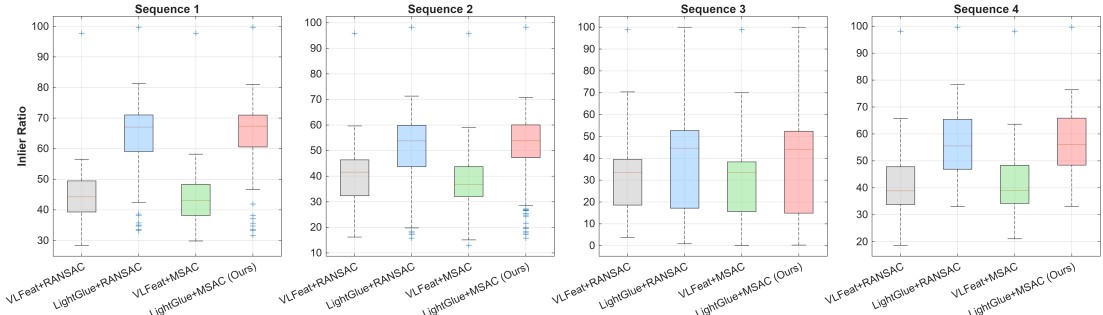

Figure 7: Ablation study on feature matching robustness. The boxplots illustrate the distribution of inlier ratios across four chosen sequences of the SCARED dataset. We compare the LightGlue + MSAC (red) against VLFeat + RANSAC (gray), LightGlue + RANSAC (blue), and VLFeat + MSAC (green). The results demonstrate that our proposed combination consistently achieves higher median inlier rates.

181.06s ($\approx 0.92s$ per frame), while pose estimation accounted for 73.15s ($\approx 0.37s$ per frame). In contrast, triangulation and pose refinement remained computationally efficient, requiring only 0.54s and 0.69s in total, respectively. This distribution indicates that while the transition to a pure Python environment with multi-CPU acceleration has streamlined the pipeline, feature matching remains the primary bottleneck, prompting further research into lightweight feature matchers and code optimization to reduce latency without compromising the robustness.

## 5. Conclusions

RVO-MIS addresses the challenges of MIS environments by integrating SIFT with Light-Glue for reliable feature correspondences, followed by advocating absolute-pose first VO framework in a robust MSAC loop. This robust combination achieves state-of-the-art performance in terms of accuracy and trajectory completion rate, both on the SCARED and EndoSLAM datasets. This suffices to show that the reliance on temporal motion consistency and photometric consistency is not entirely practical in the MIS scenarios. Notably, it requires significantly fewer computational resources than pure deep learning models, offering an alternative solution for next-generation surgical navigation.

## 6. Future Works

Noting that DPVO (Teed et al., 2023) achieved the lowest quantitative ATE ($\mathcal{T}$) in Sequence 3, we will investigate the underlying causes in future work. To further refine our system, we propose three key improvements: implementing three-view feature matching for scale-aware estimation (Yuan et al., 2017); integrating advanced matchers such as LoFTR (Sun et al., 2021) to additionally enhance estimation precision; and investigating the role of bundle adjustment (Saha et al., 2025) and its effectiveness in mitigating the motion drift.

## Acknowledgments

The authors gratefully acknowledge the support provided by the University of Arizona Graduate College through the Research and Project (ReaP) Grant.

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

## Appendix A. Algorithm Details

---

**Algorithm 1:** RVO-MIS Pipeline

---

**Input:** Sequence of video frames $I_0, \ldots, I_N$, Camera Intrinsic Matrix $K$

**Hyperparameters:**

   Inlier Threshold $\tau_{inlier} = 2$ px

   Max Iterations $N_{iter} = 3000$

   Feature Ratio $\rho_{rank} = 0.8$ (Top-k selection)

   Keyframe Gap $\tau_{gap} = 15$ frames

   Covisibility Ratio $\tau_{cov} = 0.55$

**Output:** Camera Trajectory $\{(\mathcal{R}_i, \mathcal{T}_i)\}$, 3D Map $\mathcal{M}$

   `// Initialization Phase`

**1** Extract SIFT features for $I_0$ and $I_1$ (Select top $\rho_{rank}$)

**2** Match features using LightGlue

**3** Estimate relative pose and triangulate initial 3D points to form map $\mathcal{M}$ (using $\tau_{inlier}$)

**4** Set Keyframe $I_{kf} \leftarrow I_1$

   `// Tracking Phase`

**5** **for** $i \leftarrow 2$ **to** $N$ **do**

**6**    Extract SIFT features for current frame $I_i$

**7**    Match features between $I_i$ and $I_{kf}$ using LightGlue

**8**    Identify 2D-3D correspondences (co-visible points) based on $\mathcal{M}$

      `// Absolute Pose Estimation`

**9**    Estimate initial pose $(\mathcal{R}_i, \mathcal{T}_i)$ using MSAC ($N_{iter}$ iterations, $\tau_{inlier}$ threshold)

**10**    Refine $(\mathcal{R}_i, \mathcal{T}_i)$ by minimizing the energy function $E(\mathcal{R}_i, \mathcal{T}_i)$ (Eq. 2) using Levenberg-Marquardt

      `// Map Management`

**11**    $N_{cov} \leftarrow$ ratio of co-visible 3D landmarks

**12**    $N_{gap} \leftarrow$ frame distance from $I_{kf}$

**13**    **if** $N_{cov} < \tau_{cov}$ **or** $N_{gap} > \tau_{gap}$ **then**

**14**       Compute relative pose $(\mathcal{R}_{kc}, \mathcal{T}_{kc})$ from the current and keyframe absolute poses (Eq. 7)

**15**       Triangulate new 2D matches satisfying epipolar constraints ($< \tau_{inlier}$)

**16**       Transform new points to world coordinates and add to $\mathcal{M}$

**17**       Update Keyframe $I_{kf} \leftarrow I_i$

**18**    **end**

**19** **end**

---

# Appendix B. Detailed Quantitative Results

Table B1: Detailed comparison of ATE on **Sequence 1** with *Global Alignment.* ("M" represents monocular method, "S" represents stereo method. Best results are **bold**, second best are underlined.)

| Method | ATE ($\mathcal{T}$) | | | | ATE ($\mathcal{R}$) | | | |
|---|---|---|---|---|---|---|---|---|
| | max | avg | min | RMSE | max | avg | min | RMSE |
| ORB-SLAM2 (M) (Mur-Artal and Tardós, 2017) | 2.7664 | 1.4517 | 0.5062 | 1.6543 | 3.1294 | 3.0534 | 2.9731 | 3.0534 |
| ORB-SLAM2 (S) (Mur-Artal and Tardós, 2017) | 2.4405 | 1.2036 | 0.1553 | 1.3091 | 0.3855 | 0.3502 | 0.3048 | 0.3511 |
| SD-DefSLAM (M) (Rodríguez et al., 2021) | 2.5968 | 0.9146 | 0.1828 | 1.0036 | 0.3957 | 0.3736 | 0.3479 | 0.3247 |
| BodySLAM (M) (Manni et al., 2024) | 0.9876 | 0.4107 | 0.0894 | 0.4504 | 0.3605 | 0.1786 | **0.0201** | 0.1991 |
| Endo-2DTAM (M) (Huang et al., 2025) | 5.1425 | 2.2075 | 1.0403 | 2.3290 | 3.1376 | 2.4347 | 0.3213 | 2.2410 |
| DPVO (M) (Teed et al., 2023) | 0.7202 | 0.3033 | **0.0185** | 0.3326 | 0.3853 | 0.2242 | 0.0438 | 0.2466 |
| EndoDepth (M) (Recasens et al., 2021) | 2.3421 | 1.0514 | 0.3591 | 1.1748 | 0.4253 | 0.3682 | 0.3264 | 0.3688 |
| RVO-MIS (M) | **0.5670** | **0.2813** | 0.0522 | **0.2970** | **0.0614** | **0.0522** | 0.0397 | **0.0523** |

Table B2: Detailed comparison of ATE on **Sequence 2** with *Global Alignment.* ("M" represents monocular method, "S" represents stereo method. Best results are **bold**, second best are underlined.)

| Method | ATE ($\mathcal{T}$) | | | | ATE ($\mathcal{R}$) | | | |
|---|---|---|---|---|---|---|---|---|
| | max | avg | min | RMSE | max | avg | min | RMSE |
| ORB-SLAM2 (M) (Mur-Artal and Tardós, 2017) | 2.1473 | 0.8472 | 0.4015 | 0.9912 | 3.1304 | 3.0805 | 3.0183 | 3.0798 |
| ORB-SLAM2 (S) (Mur-Artal and Tardós, 2017) | 2.5160 | 1.2091 | 0.0669 | 1.3623 | 3.1203 | 3.0298 | 2.9297 | 3.0303 |
| SD-DefSLAM (M) (Rodríguez et al., 2021) | 3.0879 | 1.6691 | 0.4306 | 1.7788 | 2.0754 | 2.0056 | 1.9468 | 2.0058 |
| BodySLAM (M) (Manni et al., 2024) | **0.9069** | 0.3935 | **0.0200** | 0.4447 | 0.2878 | 0.1749 | **0.0796** | 0.1851 |
| Endo-2DTAM (M) (Huang et al., 2025) | 5.5302 | 2.1425 | 0.3685 | 2.3187 | 3.1264 | 1.4555 | 0.7135 | 1.1858 |
| DPVO (M) (Teed et al., 2023) | 1.0712 | 0.3390 | 0.0415 | 0.3902 | 0.2489 | 0.1723 | 0.1058 | 0.1763 |
| EndoDepth (M) (Recasens et al., 2021) | 2.8358 | 1.3288 | 0.1519 | 1.4863 | 1.7598 | 1.7174 | 1.6732 | 1.7176 |
| RVO-MIS (M) | 0.9167 | **0.3090** | 0.0644 | **0.3574** | **0.1636** | **0.1285** | 0.0865 | **0.1302** |

Table B3: Detailed comparison of ATE on **Sequence 3** with *Global Alignment.* ("M" represents monocular method, "S" represents stereo method. Best results are **bold**, second best are underlined.)

| Method | ATE $(\mathcal{T})$ | | | | ATE $(\mathcal{R})$ | | | |
|---|---|---|---|---|---|---|---|---|
| | max | avg | min | RMSE | max | avg | min | RMSE |
| ORB-SLAM2 (M) (Mur-Artal and Tardós, 2017) | 21.9279 | 8.5117 | 1.2194 | 9.6247 | 3.1362 | 2.8618 | 2.6213 | 2.8640 |
| ORB-SLAM2 (S) (Mur-Artal and Tardós, 2017) | 28.0701 | 8.5828 | 4.0676 | 9.7053 | 3.1416 | 3.0071 | 2.6978 | 3.0089 |
| SD-DefSLAM (M) (Rodríguez et al., 2021) | 15.9008 | 5.5097 | 0.7359 | 6.2579 | 0.5240 | 0.3597 | 0.1703 | 0.3455 |
| BodySLAM (M) (Manni et al., 2024) | - | - | - | - | - | - | - | - |
| Endo-2DTAM (M) (Huang et al., 2025) | 30.2704 | 7.1538 | 2.1716 | 6.3792 | 3.1406 | 1.8262 | 0.1627 | 1.9767 |
| DPVO (M) (Teed et al., 2023) | **7.0309** | **2.3949** | 0.1415 | **2.8271** | 1.1005 | 0.3471 | **0.0477** | 0.4607 |
| **Deep Learning** | | | | | | | | |
| EndoDepth (M) (Recasens et al., 2021) | 31.2774 | 7.3084 | 0.4616 | 9.2599 | 0.9811 | 0.8782 | 0.6674 | 0.8804 |
| RVO-MIS (M) | 18.2302 | 2.8213 | **0.1165** | 4.0381 | **0.2317** | **0.1246** | 0.0743 | **0.1261** |

Table B4: Detailed comparison of ATE on **Sequence 4** with *Global Alignment.* ("M" represents monocular method, "S" represents stereo method. Best results are **bold**, second best are underlined.)

| Method | ATE $(\mathcal{T})$ | | | | ATE $(\mathcal{R})$ | | | |
|---|---|---|---|---|---|---|---|---|
| | max | avg | min | RMSE | max | avg | min | RMSE |
| ORB-SLAM2 (M) (Mur-Artal and Tardós, 2017) | 10.8781 | 3.6578 | 0.7318 | 4.4825 | 3.1384 | 3.0179 | 2.9487 | 3.0183 |
| ORB-SLAM2 (S) (Mur-Artal and Tardós, 2017) | 7.8057 | 3.8878 | 0.5753 | 4.3626 | 3.1405 | 2.9113 | 2.6823 | 2.9151 |
| SD-DefSLAM (M) (Rodríguez et al., 2021) | 8.9583 | 2.9963 | 0.3520 | 3.5653 | 0.3344 | 0.1665 | 0.0593 | 0.1761 |
| BodySLAM (M) (Manni et al., 2024) | 15.1282 | 7.8329 | 4.2494 | 8.2481 | 1.1025 | 0.7804 | 0.5897 | 0.7986 |
| Endo-2DTAM (M) (Huang et al., 2025) | 17.0963 | 5.0704 | 1.4731 | 5.1394 | 3.1411 | 2.6069 | 1.0978 | 2.3166 |
| DPVO (M) (Teed et al., 2023) | 1.7126 | 0.8578 | 0.1315 | 0.9269 | 0.8780 | 0.3292 | **0.0317** | 0.4405 |
| EndoDepth (M) (Recasens et al., 2021) | 8.9253 | 3.8852 | 0.6026 | 4.2948 | 0.4404 | 0.3432 | 0.2228 | 0.3484 |
| RVO-MIS (M) | **1.4768** | **0.6184** | **0.0224** | **0.6822** | **0.0778** | **0.0541** | 0.0382 | **0.0548** |

Table B5: Detailed comparison of ATE on **Sequence 1** with *Origin Alignment*. ("M" represents monocular method, "S" represents stereo method. Best results are **bold**, second best are underlined.)

| Method | ATE ($\mathcal{T}$) | | | | ATE ($\mathcal{R}$) | | | |
|---|---|---|---|---|---|---|---|---|
| | max | avg | min | RMSE | max | avg | min | RMSE |
| ORB-SLAM2 (M) (Mur-Artal and Tardós, 2017) | 10.5358 | 7.8183 | **0.0000** | 8.3241 | 0.1714 | 0.1167 | **0.0000** | 0.1259 |
| ORB-SLAM2 (S) (Mur-Artal and Tardós, 2017) | 3.7592 | 3.3326 | **0.0000** | 3.4019 | 0.0692 | 0.0493 | **0.0000** | 0.0524 |
| SD-DefSLAM (M) (Rodríguez et al., 2021) | 4.4132 | 3.0186 | **0.0000** | 3.1928 | 0.0581 | 0.0375 | 0.0004 | 0.0391 |
| BodySLAM (M) (Manni et al., 2024) | 5.2943 | 2.2945 | **0.0000** | 2.7290 | 0.6214 | 0.3839 | 0.0004 | 0.4290 |
| Endo-2DTAM (M) (Huang et al., 2025) | 15.8324 | 8.8650 | **0.0000** | 9.6275 | 3.0717 | 2.1193 | 0.0004 | 2.2411 |
| DPVO (M) (Teed et al., 2023) | 2.0840 | 0.9160 | **0.0000** | 1.1042 | 0.5907 | 0.3704 | 0.0004 | 0.4124 |
| EndoDepth (M) (Recasens et al., 2021) | 6.3598 | 2.7282 | **0.0000** | 3.0702 | 0.1119 | 0.0476 | 0.0004 | 0.0534 |
| RVO-MIS (M) | **1.3970** | **0.7528** | **0.0000** | **0.7963** | **0.0284** | **0.0172** | **0.0000** | **0.0179** |

Table B6: Detailed comparison of ATE on **Sequence 2** with *Origin Alignment*. ("M" represents monocular method, "S" represents stereo method. Best results are **bold**, second best are underlined.)

| Method | ATE ($\mathcal{T}$) | | | | ATE ($\mathcal{R}$) | | | |
|---|---|---|---|---|---|---|---|---|
| | max | avg | min | RMSE | max | avg | min | RMSE |
| ORB-SLAM2 (M) (Mur-Artal and Tardós, 2017) | 189.8000 | 79.4784 | **0.0000** | 98.5070 | 2.6856 | 1.2635 | **0.0000** | 1.4628 |
| ORB-SLAM2 (S) (Mur-Artal and Tardós, 2017) | 17.2767 | 4.2766 | **0.0000** | 5.9156 | 0.3030 | 0.0859 | **0.0000** | 0.1176 |
| SD-DefSLAM (M) (Rodríguez et al., 2021) | 10.5063 | 0.7037 | **0.0000** | 1.3217 | 0.1687 | **0.0182** | **0.0000** | 0.0319 |
| BodySLAM (M) (Manni et al., 2024) | 1.8654 | 0.7496 | **0.0000** | 0.8442 | 0.3314 | 0.1824 | 0.0015 | 0.2093 |
| Endo-2DTAM (M) (Huang et al., 2025) | 5.0698 | 2.8493 | **0.0000** | 3.0723 | 3.1312 | 0.6630 | 0.0015 | 1.1861 |
| DPVO (M) (Teed et al., 2023) | 1.2917 | **0.5071** | **0.0000** | 0.5583 | 0.2033 | 0.1134 | 0.0015 | 0.1252 |
| EndoDepth (M) (Recasens et al., 2021) | 3.3008 | 1.7438 | **0.0000** | 1.8933 | 0.0733 | 0.0345 | 0.0015 | 0.0386 |
| RVO-MIS (M) | **1.2284** | 0.5072 | **0.0000** | **0.5528** | **0.0419** | 0.0202 | **0.0000** | **0.0236** |

Table B7: Detailed comparison of ATE on **Sequence 3** with *Origin Alignment*. ("M" represents monocular method, "S" represents stereo method. Best results are **bold**, second best are underlined.)

| Method | ATE ($\mathcal{T}$) | | | | ATE ($\mathcal{R}$) | | | |
|---|---|---|---|---|---|---|---|---|
| | max | avg | min | RMSE | max | avg | min | RMSE |
| ORB-SLAM2 (M) (Mur-Artal and Tardós, 2017) | 425.3198 | 198.6521 | **0.0000** | 234.2704 | 31.5492 | 15.2104 | 0.0017 | 17.9475 |
| ORB-SLAM2 (S) (Mur-Artal and Tardós, 2017) | 51.5513 | 23.1523 | **0.0000** | 27.8263 | 1.2640 | 0.4475 | 0.0017 | 0.5553 |
| SD-DefSLAM (M) (Rodríguez et al., 2021) | 16.5036 | 7.7565 | **0.0000** | 8.5289 | **0.3199** | 0.1463 | 0.0017 | 0.1610 |
| BodySLAM (M) (Manni et al., 2024) | - | - | - | - | - | - | - | - |
| Endo-2DTAM (M) (Huang et al., 2025) | 38.1482 | 11.1089 | **0.0000** | 13.2626 | 3.1412 | 2.0323 | 0.0017 | 2.1547 |
| DPVO (M) (Teed et al., 2023) | **9.7135** | 2.7100 | **0.0000** | **3.5819** | 1.2508 | 0.4057 | 0.0017 | 0.5303 |
| EndoDepth (M) (Recasens et al., 2021) | 36.6869 | 7.8943 | **0.0000** | 10.4578 | 0.7332 | 0.1893 | 0.0017 | 0.2422 |
| RVO-MIS (M) | 22.1006 | **2.6989** | 0.0000 | 4.7195 | 0.3386 | **0.0522** | **0.0000** | **0.0830** |

Table B8: Detailed comparison of ATE on **Sequence 4** with *Origin Alignment*. ("M" represents monocular method, "S" represents stereo method. Best results are **bold**, second best are underlined.)

| Method | ATE ($\mathcal{T}$) | | | | ATE ($\mathcal{R}$) | | | |
|---|---|---|---|---|---|---|---|---|
| | max | avg | min | RMSE | max | avg | min | RMSE |
| ORB-SLAM2 (M) (Mur-Artal and Tardós, 2017) | 210.3456 | 103.1932 | **0.0000** | 128.9915 | 3.1415 | 1.5821 | 0.0011 | 1.9776 |
| ORB-SLAM2 (S) (Mur-Artal and Tardós, 2017) | 28.5123 | 13.7664 | **0.0000** | 17.2080 | 0.4512 | 0.2185 | 0.0011 | 0.2732 |
| SD-DefSLAM (M) (Rodríguez et al., 2021) | 6.1234 | 3.0819 | **0.0000** | 3.8524 | 1.1023 | 0.5299 | 0.0011 | 0.6624 |
| BodySLAM (M) (Manni et al., 2024) | 24.1587 | 9.9346 | **0.0000** | 11.2834 | 0.7063 | 0.3807 | 0.0011 | 0.4392 |
| Endo-2DTAM (M) (Huang et al., 2025) | 20.9071 | 7.1370 | **0.0000** | 8.1023 | 3.1410 | 2.1662 | 0.0011 | 2.3157 |
| DPVO (M) (Teed et al., 2023) | 4.1829 | 2.1652 | **0.0000** | 2.3902 | 0.9751 | 0.3931 | 0.0011 | 0.5074 |
| EndoDepth (M) (Recasens et al., 2021) | 10.9638 | 5.8307 | **0.0000** | 6.4815 | 0.1662 | 0.0930 | 0.0011 | 0.1013 |
| RVO-MIS (M) | **2.1564** | **1.3026** | 0.0000 | **1.3573** | **0.0512** | **0.0218** | **0.0000** | **0.0237** |

