# OpenReview forum: "RVO-MIS: Robust Visual Odometry for Minimally Invasive Surgery"
_MIDL.io/2026/Conference — MIDL 2026 Poster_

### Official Review · Reviewer_ifp7 · 2026-01-06

**Confidence:** 4
**Preliminary Rating:** 4

**Summary:**

Authors present a new visual odometry method for minimally invasive
surgery (MIS) scenarios. Accurate estimation of camera pose in MIS is
crucial for various application. In the recent years, method using
deep learning based depth estimation have been extensively used for
VO. Authors here note that inaccurate depth estimation can lead to
inaccurate came pose estimation. Traditional feature-based methods
also face signficant challenges, especially due to reflections and
other artifacts in MIS scenarios. Instead, authors propose SIFT
feature extraction and DL-based matching in this work. Once the
matches are obtained, an established method for identifying external
relative camera parameters are used. Keyframes are used for the
relative positioning and updated based on criteria defined by the
authors. Experiments on the SCARED dataset are presented.

**Strengths:**

+ The method relies on well established methodology and justified
  design choices. Through the DL-based matching, authors were able to
  make good old feature-based reconstruction work very well.
+ Results in Table 1 are very good. They support the method.
+ The article is well written, design choices are well justified, and the
  experiments are clearly explained.

**Weaknesses:**

- The level of technical novelty is not very high. Authors rely on
  well-established feature matching methods for VO. The main novelty
  is the use of an existing method, Lightglue.
- Comparisons between using Lightglue and VLFeat would have been very
  nice. This comparison would have highlighted the value of the
  DL-based matching.

**Detailed Comments:**

Please see the weaknesses section.

I think the main weakness is the level of novelty, however, there is not much
authors can do to that end.

**Justification Of The Preliminary Rating:**

The article does not present a very novel technique. However, it is presenting a modification of a
well established approach with a novel component. In doing so, authors demonstrate that the
well established method can indeed reach very accurate VO estimates. This is an important point.
Experimental evaluation is also solid in my opinion.

**Questions To Address In The Rebuttal:**

My main question would be on the comparison between using Lightglue and VLFeat.
Quantifying the improvement due to using Lightglue in the presented framework would
be a valuable contribution.

---

> ### Author Response · Authors · 2026-01-25
>
> We sincerely thank the reviewer  for the positive assessment of our work and for recognizing the effectiveness of our design choices in achieving high-accuracy VO. We particularly appreciate the suggestion to quantitatively compare LightGlue with VLFeat; we have performed this ablation study to explicitly demonstrate the significant improvements contributed by deep learning-based matching.
>
> **Comment 1:** The level of technical novelty is not very high. Authors rely on well-established feature matching methods for VO. The main novelty is the use of an existing method, Lightglue.
>
> **Response:** Our contribution lies in reformulating visual odometry as a sequence of robust absolute pose estimation problems with respect to a 3D map with an internally consistent scale, rather than as an incremental relative motion estimation or a temporally coupled reprojection-error minimization process. Specifically, our novelties are threefold: (i) as opposed to local reprojection refinement typically adopted by the existing methods, camera poses are estimated directly with respect to a fixed world coordinate system using 2D–3D correspondences and MSAC-based PnP, making absolute pose recovery the core estimation objective; (ii) robustness to outliers is treated as a first-class component of the estimator: pose hypotheses are selected through MSAC over geometric constraints, with least-squares optimization applied only as a lightweight refinement on inlier supports, rather than as the dominant estimation mechanism; (iii) the proposed framework integrates SIFT feature detection in conjunction with LightGlue feature matching to provide reliable feature correspondence in support of robust pose estimation.
> We recognized that the original manuscript did not make the novelties clear, and thus a list of contributions have been made at the end of the introduction section in the updated version.
>
> **Comment 2:** Comparisons between using Lightglue and VLFeat would have been very nice. This comparison would have highlighted the value of the DL-based matching.
>
> **Response:** We have conducted an ablation study comparing the inlier ratio and accuracy using four settings from the combination of using VLFeat or Lightglue and using RANSAC or MSAC. As shown in the new Table 3 and Figure 7, the quantitative results demonstrate that replacing VLFeat with LightGlue significantly enhances the system's robustness. Specifically, the use of LightGlue increases the average Inlier Ratio from $\sim$38% to $\sim$52% and improves the Average Trajectory Completion Rate from $\sim$82% to 100% across the SCARED and EndoSLAM datasets. This confirms that deep learning-based matching effectively prevents tracking failures in challenging MIS scenarios where traditional descriptors often fail. Furthermore, our proposed combination (LightGlue + MSAC) achieves the lowest pose estimation error (Translation ATE: 2.03 cm vs. 2.73 cm for VLFeat + MSAC), validating the critical contribution of this component to the overall accuracy.

---

### Official Review · Reviewer_Wjw8 · 2026-01-10

**Confidence:** 4
**Preliminary Rating:** 1

**Summary:**

This paper proposes RVO-MIS, a monocular visual odometry pipeline for minimally invasive surgery that avoids depth-based photometric VO and instead relies on geometric estimation with robust optimization to handle specularities, smoke, low texture, and tool-induced outliers. The method detects keypoints with SIFT and matches them using LightGlue, initializes a sparse 3D map by triangulating early-frame correspondences, and then estimates each new camera pose ($(R_i,T_i)$) from 2D–3D correspondences using MSAC with P3P/PnP followed by Levenberg–Marquardt refinement that minimizes reprojection error. Keyframes are inserted using simple visibility/time heuristics, and new landmarks are added via triangulation to grow the map over time. Experiments on selected SCARED sequences report absolute trajectory error (ATE) after global Sim(3) alignment and compare against several VO/SLAM baselines, suggesting improved robustness in some runs; the main significance is an interpretable, feature-and-geometry-driven alternative to depth-supervised MIS VO, though the practical strength of the evidence depends on baseline configurations and the limited dataset coverage.

**Strengths:**

1. The work presents a well-motivated "systems" integration that bridges classical geometry with modern deep learning. By combining SIFT for reliable keypoint extraction with LightGlue for robust matching under surgical nuisance factors, the authors provide a modular framework that is significantly easier to adapt and audit than end-to-end black-box models.

2. Unlike many recent deep-learning-only approaches, this method maintains a clear, mathematically sound core. The use of standard components, including MSAC, P3P, and explicit reprojection-error refinement via Levenberg-Marquardt optimization. It ensures the pose estimation process remains transparent and computationally efficient compared to dense photometric alternatives.

3. The paper provides a compelling scientific justification for moving away from depth-supervised photometric tracking. The authors correctly identify that the specular reflections and unpredictable illumination common in surgical scenes frequently violate photometric assumptions. Their focus on geometric optimization with robust consensus is a sensible and effective direction for high-stakes clinical navigation.

4. The experimental section is thorough and standard evaluation protocols like ATE with $Sim(3)$ alignment. Furthermore, the detailed runtime analysis, which identifies feature matching as the primary computational bottleneck, provides valuable "real-world" insight for practitioners looking to optimize or deploy similar navigation stacks.

**Weaknesses:**

1. **Limited Experimental Scope and Narrow Validation:**
The evaluation is primarily confined to the SCARED dataset, specifically targeting rigid porcine scenes without respiratory motion. While the paper claims robustness to MIS challenges, the absence of data featuring non-rigid tissue deformation or broader intraoperative variability weakens the generalizability of these claims. A more comprehensive study would require testing on additional datasets or systematically stress-testing the pipeline against the stated challenges of surgical smoke and severe specularities.

2. **Ambiguity in Scale Recovery and Evaluation Protocol:**
There is a discrepancy between the method's motivation and its evaluation. The authors describe a triangulation-based initialization designed to "fix" the metric scale in the first-frame coordinate system. However, the reported Absolute Trajectory Error (ATE) is computed using a $Sim(3)$ alignment that optimizes for scale. This makes it difficult to ascertain if the system successfully recovers true metric scale or remains "up-to-scale," necessitating an evaluation without scale alignment to verify scientific accuracy.

3. **Insufficient Ablation of the Matching Component:** A core premise of the work is that LightGlue provides superior robustness over conventional descriptor-based matching in MIS environments. Despite this, the paper lacks a rigorous ablation study comparing the proposed SIFT+LightGlue configuration against standard baselines (e.g., SIFT with ratio tests) or alternative modern matchers. Quantifying exactly when and where LightGlue fails during challenging sequences, such as those with high specularity or low contrast, is essential to understanding the method's robustness envelope.

4. **Underspecified Implementation Details and Reproducibility:** While certain thresholds are provided, such as the 2-pixel epipolar constraint and keyframe triggers based on co-visibility (<0.55) or frame distance (>15), several critical hyperparameters remain underspecified. Details regarding the number of retained keypoints, specific MSAC inlier thresholds, and map-pruning strategies are not clearly consolidated, which may hinder the reproducibility of the results.

5. **Computational Bottlenecks and Real-Time Feasibility:** The runtime analysis reveals that the system is currently not compatible with real-time surgical use, with feature matching alone requiring $\approx0.92$s per frame and pose recovery adding $\approx0.37$s. While this transparency is appreciated, the paper would be strengthened by a speed-accuracy trade-off study to determine if lightweight matchers or reduced keypoint counts could bring the pipeline closer to real-time operation without significant performance degradation.

**Detailed Comments:**

1. **Clarify and enumerate contributions early.** The paper would be easier to evaluate if the introduction (or end of Section 1) explicitly listed 2–4 concrete contributions , rather than leaving readers to infer novelty from the method description.

2. **Improve the structure of the Methods section.** Section 3 currently reads as a collection of narrow subsections that do not always build a clean narrative from inputs ($\rightarrow$) correspondences ($\rightarrow$) pose estimation ($\rightarrow$) keyframing (\rightarrow) map update. A small restructuring suggestion:

  * 3.1 Overview (one paragraph + pipeline figure/pseudocode)
  * 3.2 Feature extraction & matching (SIFT + LightGlue settings)
  * 3.3 Initialization (relative pose + triangulation + thresholds)
  * 3.4 Tracking (pose estimation) (MSAC-P3P/PnP + LM refinement + inlier criteria)
  * 3.5 Mapping / keyframe management (visibility/time triggers, triangulation, pruning)
    This would reduce repetition and make it much clearer which parts are novel vs standard.

3. **Figure 3: make the comparison more visually discriminative.** The current trajectory visualization is difficult to read for “which method is better” at a glance, especially when paths overlap. Consider adding:

  * Projected views (e.g., (x)-(y), (x)-(z), or (y)-(z) projections) alongside the 3D plot to highlight deviations more clearly.
  * Error overlays (e.g., per-frame color-coded translation error, or a separate plot of error vs. frame index) to show where drift accumulates and which segments drive ATE differences.
  * Clearer legend and consistent alignment choice (origin-aligned vs globally aligned), or explicitly label which alignment is used for each plot to avoid confusion.

**Justification Of The Preliminary Rating:**

I selected strong reject mainly because the current submission falls slightly below the bar for a weak reject, but the available rating options do not allow a finer gradation. The core idea, combining SIFT keypoints, LightGlue matching, and MSAC-PnP pose estimation with triangulation-based map growth, is reasonable and could serve as an engineering baseline.  However, the paper’s writing and method exposition are not yet strong enough: Section 3 is fragmented and underspecified, making it hard to follow the end-to-end logic and difficult to reproduce the system beyond the high-level pipeline.  The experimental evidence is also too limited (primarily SCARED, with rigid scenes noted by the authors), and it does not provide a systematic worst-case analysis for the challenges emphasized in the motivation (specularities, smoke, low contrast, tool motion).  Moreover, the narrative around “fixing scale” via initial triangulation is not fully consistent with reporting ATE after Sim(3) alignment (which optimizes scale during evaluation), weakening the interpretation of metric-scale claims.

Overall, I see promise, but the paper would need clearer methodological presentation, stronger ablations (especially isolating the matcher’s contribution), and broader/more diagnostic experiments before it is ready for publication.

**Questions To Address In The Rebuttal:**

1. **What is the true scale behavior of RVO-MIS?**
The manuscript states that the initialization phase reconstructs 3D points in the first frame's coordinate system to "fix" the scale for subsequent absolute pose estimation. However, the reported Absolute Trajectory Error (ATE) is calculated after a  alignment that optimizes for rotation, translation, and scale.

* Does the system produce metrically-scaled trajectories without utilizing ground truth information at any stage?
* If the system is inherently metric, what serves as the source of metric scale in this monocular setup, and can you provide ATE results under alignment (rotation and translation only) to demonstrate scale consistency?

2. **To what extent is the performance gain attributable specifically to LightGlue versus other pipeline components?**
The authors identify conventional descriptor-based matching as a primary failure point in MIS environments due to ambiguous features.

* Can you provide a focused ablation study comparing the proposed SIFT + LightGlue configuration against standard baselines (e.g., SIFT + ratio test/bidirectional matching) using the same detector?
* Please include metrics such as inlier ratios, average track length, and tracking failure rates, particularly during segments with high specularity or smoke.


3. **How does the system perform under true "worst-case" conditions and across a broader variety of sequences?**
The paper motivates the need for robustness against surgical smoke, reflections, and tool movements, yet the current evaluation is limited to rigid porcine scenes in the SCARED dataset.

* Can you provide a targeted analysis (both qualitative and quantitative) on sequences where baselines typically fail, such as those with significant respiratory motion or dense smoke, to demonstrate if RVO-MIS fails more gracefully?
* To support the claim of "robustness," would it be possible to report the failure rate across a larger subset of the dataset to ensure the current results are not localized to a few select sequences?

4. **Table 1: clarify baseline configurations and why some errors are extremely large (especially Sequences 3–4).**

* Table 1 reports translation ATE (RMSE) in cm and rotation error in degrees, after Sim(3) alignment (scale + rotation + translation) for all monocular methods. Even with Sim(3) alignment, several baselines show very large translation errors in Sequence 3 and 4.
* This raises a reproducibility/fairness question: were all comparison methods run using their official pretrained models and recommended parameters for SCARED, or retrained/tuned on SCARED? If not retrained, please state this explicitly and report key configuration details (input resolution, frame rate/subsampling, initialization strategy, and failure-handling). If retrained, please specify the training split and supervision used. Without this, it is difficult to interpret whether the “significant outperformance” reflects method advantages or baseline misconfiguration / domain mismatch.

---

> ### Author Response · Authors · 2026-01-25
>
> **Comment 4:** Underspecified implementation details and reproducibility.
>
> **Response:** We have updated the manuscript by adding more details in the selection of hyper-parameters, e.g., inlier threshold and maximal number of iterations in MSAC. To benefit the community, we will release the code upon acceptance in GitHub (https://github.com/vsi-lab/RVOMIS).
>
> **Comment 5:** Table 1: clarify baseline configurations and why some errors are extremely large (especially Sequences 3–4).
>
> **Response:** The competing baselines are all evaluated using their default settings, namely, for learning-based methods such as DPVO and EndoDepth, their pretrained weights are used. In addition, the input resolution for all methods is the native resolution from the dataset without downsampling. The corresponding strategies for failure handling and initialization are now introduced in the related works and emphasized in the experiments for failure demonstration in the experiment section of the updated manuscript.
> The inferior performances from the baseline methods shown in Table 1 are explained and justified as follows. Methods such as ORB-SLAM2 and SD-DefSLAM estimate camera poses by first giving an initial guess from a motion-prior, typically based on a constant motion assumption, and then refine the initial guess via minimizing the local reprojection errors. The situation in the MIS environment, however, typically exhibits abrupt motion that easily breaks the assumption. Since minimizing reprojection errors is not a convex optimization process, a poor initial guess would converge to a local optima, leading to inaccurate estimation. For methods such as BodySLAM, camera pose estimation is tightly coupled with non-rigid shape optimization and relies on both temporal coherence and model-based constraints to stabilize tracking. The strong parametric prior for modeling the deformable shape acts like an over-constrained optimization that is typically biased, especially in the unpredictable, changing MIS environment. The reliance on temporal coherence makes the system vulnerable to abrupt camera motions, a limitation shared by Endo-2DTAM. Specifically, Endo-2DTAM employs a constant velocity motion model for pose initialization, which fails to predict the camera's state accurately during rapid acceleration or erratic movements, leading to tracking divergence. Furthermore, baselines like DPVO and EndoDepth, heavily rely on short-term photometric consistency. This assumption is frequently violated in MIS environments due to the collocated camera-light setup, where illumination changes significantly with camera movement, and specular reflections from fluids create 'virtual' motions that contradict the scene geometry.
>
> **Comment 6:** Computational bottlenecks and real-time feasibility.
>
> **Response:** We appreciate the reviewer's emphasis on real-time feasibility for intraoperative use. To clarify, our main goal is robustness to common MIS challenges, and we report runtime mainly for transparency. As our results show, the dominant cost comes from feature matching, while pose recovery is comparatively lightweight. We will make this clearer in the revised manuscript and explicitly note that the current implementation is not yet real-time, listing it as future work. That said, this computational cost stems from the current matching configuration, not from the geometric formulation itself, and can be reduced through engineering choices and parallel implementation, such as limiting the keypoint budget, downsampling, ROI masking, or lighter matchers, without changing the core estimation pipeline.
>
> **Comment 7:** Clarify and enumerate contributions early. Improve the structure of the Methods section. Make the comparison (Figure 3) more visually discriminative.
>
> **Response:** We appreciate the valuable suggestions on making the paper clear and well-organized. We have addressed these points in the revised manuscript. Specifically, (i) a list of novelties and contributions have been included at the end of the introduction section; (ii) subsections under Section 3 have been reordered, with an updated, nicely drawn flow chart of Figure 2 assisting the reader follow the proposed framework; (iii) trajectories along each dimension are presented in Figure 5 to better demonstrate the performances across different methods.

---

> ### Author Response · Authors · 2026-01-25
>
> We express our sincere gratitude to the reviewer for the exceptionally detailed and constructive critique. Your rigorous analysis particularly regarding the scale ambiguity, experimental scope, and the structure of Section 3 has been invaluable in guiding the extensive revisions of our manuscript.
>
> **Comment 1:** Limited Experimental Scope and Narrow Validation. How does the system perform under true "worst-case" conditions and across a broader variety of sequences?
>
> **Response:** We have extended the experiment section in the updated manuscript in threefold: (i) detailed descriptions of the chosen sequences from the SCARED dataset are given, with example pictures in Figure 3 showing the challenging situations present in the dataset; (ii) an additional dataset, EndoSLAM, which exhibits drastic camera motion attributing to blurry images, is involved for extended evaluations; (iii) an ablation study on the EndoSLAM dataset is provided to show how RVO-MIS performs in the very poor cases.
>
>
> **Comment 2:** [There is an] ambiguity in scale recovery and evaluation protocol. What is the true scale behavior of RVO-MIS?
>
> Response: Since the input to the proposed framework is a sequence of monocular images, recovering the true metric scale of the scene and camera motion is fundamentally impossible. Rather, what can be achieved is the fixation of an “internally consistent scale”. By triangulating 3D points from relative motion estimates, a particular yet arbitrary scale is implicitly chosen for the reconstructed 3D structure. Subsequent absolute camera pose estimations are then computed with respect to these triangulated points, enforcing scale consistency across all estimated poses. This enables a trajectory to be expressed in a single, “fixed” internal scale, without the knowledge of the true metric scale of the real world. When comparing the estimated trajectory to a ground-truth trajectory expressed in metric units, a Sim(3) alignment is required to account for the global scale ambiguity for a meaningful evaluation. We recognized that the manuscript may provide vague descriptions on the “scale”, and thus it has been addressed and clarified in the updated paper.
>
> **Comment 3:** Insufficient ablation of the matching component. To what extent is the performance gain attributable specifically to LightGlue versus other pipeline components?
>
> **Response:** We have augmented the experiment section in the updated manuscript to include the ablation study by varying the choice of feature matching method (VLFeat versus LightGlue) and the geometric verification combinations (RANSAC versus MSAC). Using the EndoSLAM dataset, the use of LightGlue in conjunction with MSAC gives the best results in terms of inlier ratio, average trajectory completion rate, and ATE.

---

### Official Review · Reviewer_vLzU · 2026-01-13

**Confidence:** 4
**Preliminary Rating:** 4
**Final Rating:** 4

**Summary:**

This research propose the RVO-MIS which addresses MIS visual odometry challenges (texturelessness, reflections, etc.) via SIFT feature extraction, LightGlue matching, keyframe triangulation to resolve scale ambiguity, and P3P+MSAC for robust pose estimation. Evaluated on the SCARED dataset, it achieves a best translation ATE (RMSE) of 0.2970 cm, outperforming state-of-the-art SLAM, VO, and depth-based deep learning methods. Without scene-specific fine-tuning, it balances accuracy and efficiency, providing a solution for surgical navigation and downstream 3D reconstruction tasks.

**Strengths:**

1. The proposed method targets MIS-specific challenges via integrating SIFT+LightGlue+P3P/MSAC, avoiding depth errors.
2. The method requires no scene-specific fine-tuning, leveraging pre-trained weights to balance accuracy and efficiency.

**Weaknesses:**

1. The study’s framework is well-detailed, but the validation of its robustness remains insufficient. It is recommended to conduct comparative experiments using datasets such as EndoSLAM (Ozyoruk et al., 2021) to further corroborate its performance across diverse MIS scenarios.
2. More comparisons with the most recent Visual Odometry algorithms (e.g., ColVO (Liu et al., 2024)) should be added to better demonstrate the proposed method’s advancement over existing approaches.
3. It is suggested to supplement the experimental section with analyses under extreme conditions such as illumination changes and severe tissue deformation, so as to explicitly verify the model’s robustness.

Ref:
Ozyoruk, Kutsev Bengisu, et al. "EndoSLAM dataset and an unsupervised monocular visual odometry and depth estimation approach for endoscopic videos." Medical image analysis 71 (2021): 102058.
Liu, Ruyu, et al. "Colvo: Colonoscopic visual odometry considering geometric and photometric consistency." Proceedings of the 32nd ACM international conference on multimedia. 2024.

**Detailed Comments:**

Please check the Weaknesses part.

**Justification Of Final Rating:**

Thanks for the response, most of my concerns have been addressed. Although the article has some room for improvement in innovativeness, the overall content still has some novelties. Based on the author’s response, my final rating is weak accept.

**Justification Of The Preliminary Rating:**

While the article is presented in a relatively clear manner, comparisons involving the new method and validations across different datasets are still lacking; thus, further revisions are recommended to substantiate the authors’ claims.

**Questions To Address In The Rebuttal:**

More comprehensive comparisons across the datasets and competing methods are needed.

---

> ### Author Response · Authors · 2026-01-25
>
> We sincerely thank the reviewer for the positive assessment of our work and the constructive comments, particularly the suggestions to extend our evaluation to the EndoSLAM dataset and compare against recent methods like ColVO, which have significantly strengthened our manuscript.
>
> **Comment 1**: The study’s framework is well-detailed, but the validation of its robustness remains insufficient. It is recommended to conduct comparative experiments using datasets such as EndoSLAM (Ozyoruk et al., 2021) to further corroborate its performance across diverse MIS scenarios.
>
> **Response**: The sequences chosen from the SCARED dataset to evaluate the proposed method and the baselines cover a diverse set of MIS conditions, including the presence of surgical instruments, flowing blood, textureless areas, and light reflections. These sequences are believed to sufficiently demonstrate the estimation robustness. We agree that other datasets such as EndoSLAM could serve to showcase the effectiveness of the methods. However, we recognized that EndoSLAM has a serious frame-skipping problem, namely, the sequence of images is not consecutive, often giving non-overlap in scene between adjacent images. As explicitly attributed by (Deng et al. 2023), exceptionally large VO errors on EndoSLAM are observed, undermining the validity of performance metrics on this dataset. That being said, EndoSLAM provides a good source to evaluate the performances under drastic camera motion and the attributed blurry images. We thus conducted additional experiments using one EndoSLAM sequence, and the quantitative results are given in Table 2 of the updated manuscript. At the time of this writing, we are including only one sequence of the EndoSLAM dataset due to limited time frame, yet we are planning to include more sequences in the camera-ready version upon acceptance.
>
> [Deng et al. 2023] Deng, Jianning, Peize Li, Kevin Dhaliwal, Chris Xiaoxuan Lu, and Mohsen Khadem. "Feature-based visual odometry for bronchoscopy: A dataset and benchmark." In 2023 IEEE/RSJ International Conference on Intelligent Robots and Systems (IROS), pp. 6557-6564. IEEE, 2023.
>
> **Comment 2**: More comparisons with the most recent Visual Odometry algorithms (e.g., ColVO (Liu et al., 2024)) should be added to better demonstrate the proposed method’s advancement over existing approaches.
>
> **Response**: We recognized that there are recent VO algorithms that appear to work well in the MIS environments, but the code cannot be sourced publicly. Specifically, the suggested method, ColVO, provides only the pretrained weights rather than the code for the model. This has been reported in its issue thread: https://github.com/HNUicda/CoIVO/issues/1, and it remains unresolved at the time of this writing. Others either provide invalid GitHub links, e.g., EndoMODE (Shao et al. 2025), or with no publicly available codebase, e.g., CudaSIFT-SLAM (Elvira et al. 2024), (Ye et al. 2025), (Huang et al. 2025), etc.
>
> [Shao et al. 2025] Shao, Liangjing, Benshuang Chen, Shuting Zhao, and Xinrong Chen. "EndoMODE: A Multimodal Visual Feature-Based Ego-Motion Estimation Framework for Monocular Odometry and Depth Estimation in Various Endoscopic Scenes." IEEE Transactions on Industrial Informatics (2025).
>
> [Elvira et al. 2024] Elvira, Richard, Juan D. Tardós, and José MM Montiel. "CudaSIFT-SLAM: multiple-map visual SLAM for full procedure mapping in real human endoscopy." arXiv preprint arXiv:2405.16932 (2024).
>
> [Ye et al. 2025] Ye, Ke, Bai Chen, Jingyang Zhou, Jiahao Li, Haoqing Wu, Feng Ju, and Yang Wu. "Enhanced visual SLAM for surgical robots with cylindrical scene recognition in digestive endoscopic procedures." Measurement 250 (2025): 117054.
>
> [Huang et al. 2025] Huang, Junjun, Tianran He, Juan Xu, Weiting Wu, and Wei Wu. "Automatic Endoscopic Navigation for Monocular Depth and Ego-Motion Estimation in Wireless Capsule Endoscopy Through Transformer Network." IEEE Access (2025).
>
> **Comment 3**: It is suggested to supplement the experimental section with analyses under extreme conditions such as illumination changes and severe tissue deformation, so as to explicitly verify the model’s robustness.
>
> **Response**: We have augmented more descriptions in the updated manuscript on the chosen sequences from the SCARED dataset exhibiting challenging situations in light reflections, flow blood, presence of surgical instruments, etc. In addition, abrupt camera motion provided by the EndoSLAM dataset is used to additionally evaluate the proposed method against the competing baselines. The results are now shown in Table 2 of the updated manuscript.

---

### Author Rebuttal · Authors · 2026-01-25

**Rebuttal:**

We have updated the latest revision of our manuscript. This version includes additional experiments and analyses aimed at addressing the reviewers' feedback on robustness and cross-dataset evaluation.

Summary of Key Changes:

New Evaluation on EndoSLAM Dataset: To validate performance across diverse MIS scenarios, we conducted additional experiments on the challenging EndoSLAM dataset (specifically the Small Intestine scene), which features rapid motion and low texture. As presented in the new Table 2, RVO-MIS achieves 100% trajectory completion and superior pose accuracy (GA-ATE and OA-ATE), whereas widely used baselines (e.g., ORB-SLAM2, SD-DefSLAM) and some learning-based methods suffer from tracking failures (drift or loss of lock).

Robustness Analysis & Ablation Study: We added a detailed quantitative ablation study (Table 3) and inlier ratio distribution analysis (Figure 7) to explicitly verify the model's robustness under extreme conditions. The results demonstrate that our proposed integration of SIFT, LightGlue, and MSAC consistently yields higher inlier rates and lower trajectory error compared to alternative configurations (e.g., using RANSAC or traditional descriptors).

Comparison with SOTA Methods: We extended our comparative analysis to include recent state-of-the-art approaches (e.g., DPVO, Endo-2DTAM). The revised results highlight our method's ability to balance accuracy and efficiency without requiring scene-specific fine-tuning.

We believe these revisions strongly substantiate the robustness and generalizability of RVO-MIS. We thank the reviewers for their constructive feedback which has significantly improved the quality of this work.

**Supporting Material:**

/attachment/daa989421b11c93f9171aebb49a7981fc0302bf5.pdf

---

### Meta-Review · Area_Chair_1vP3 · 2026-02-07

**Recommendation:** Accept (Poster)
**Confidence:** 5

**Metareview:**

The revised paper during the rebuttal period addressed most of the comments, with many changes, especially for the validation, as the reviewers suggested. I think the current content may reach the bar. However, as reviewers mentioned, novelty is limited.

---

### Decision · Program_Chairs · 2026-02-13

Accept (Poster)